# Molecular and Genetic Mechanisms of Spinal Stenosis Formation: Systematic Review

**DOI:** 10.3390/ijms232113479

**Published:** 2022-11-03

**Authors:** Vadim A. Byvaltsev, Andrei A. Kalinin, Phillip A. Hernandez, Valerii V. Shepelev, Yurii Y. Pestryakov, Marat A. Aliyev, Morgan B. Giers

**Affiliations:** 1Department of Neurosurgery, Irkutsk State Medical University, Irkutsk 664003, Russia; 2Department of Traumatology, Orthopedic and Neurosurgery, Irkutsk State Medical Academy of Postgraduate Education, Irkutsk 664049, Russia; 3School of Chemical, Biological, and Environmental Engineering, Oregon State University, Corvallis, OR 97331, USA; 4Department of Neurosurgery, Asfendiyarov Kazakh National Medical University, Almaty 050012, Kazakhstan

**Keywords:** spinal stenosis, genetics, molecular mechanisms, degenerative disease, congenital disease, systematic review

## Abstract

Spinal stenosis (SS) is a multifactorial polyetiological condition characterized by the narrowing of the spinal canal. This condition is a common source of pain among people over 50 years old. We perform a systematic review of molecular and genetic mechanisms that cause SS. The five main mechanisms of SS were found to be ossification of the posterior longitudinal ligament (OPLL), hypertrophy and ossification of the ligamentum flavum (HLF/OLF), facet joint (FJ) osteoarthritis, herniation of the intervertebral disc (IVD), and achondroplasia. FJ osteoarthritis, OPLL, and HLF/OLFLF/OLF have all been associated with an over-abundance of transforming growth factor beta and genes related to this phenomenon. OPLL has also been associated with increased bone morphogenetic protein 2. FJ osteoarthritis is additionally associated with Wnt/β-catenin signaling and genes. IVD herniation is associated with collagen type I alpha 1 and 2 gene mutations and subsequent protein dysregulation. Finally, achondroplasia is associated with fibroblast growth factor receptor 3 gene mutations and fibroblast growth factor signaling. Although most publications lack data on a direct relationship between the mutation and SS formation, it is clear that genetics has a direct impact on the formation of any pathology, including SS. Further studies are necessary to understand the genetic and molecular changes associated with SS.

## 1. Introduction

Spinal stenosis (SS) is a multifactorial polyetiological condition characterized by the narrowing of the spinal canal, leading to spinal cord compression [1]. This condition is a common source of pain and decreased function among people over 50 years old [2]. Lumbar SS causes morbidity requiring surgery in 14 out of every 10,000 people over 65 in the US [3]. Up to 3.9% of patients seeking care for low back pain have SS [4]. SS is classified by etiology and anatomical location. Stenosis etiology can be primary (congenital or acquired) or secondary (caused by other diseases). Anatomically, stenosis can be central or foraminal, single or multilevel, isolated or tandem [5]. In this article, we review primary SS and its molecular and genetic causes. The authors believe that a deeper understanding of the genetic and molecular mechanisms of stenosis formation will help catalyze the creation of novel therapies for SS patients. Thus, we also outline areas that require further study by the spine community.

## 2. Methods

A systematic review of genetic mutations associated with SS was performed. Searches were conducted in the PubMed, CNKI, Cochrane Library, and eLibrary databases for articles published between January 1990 and April 2021. Data were obtained by two authors and independently reviewed by a third author. Disagreements regarding study inclusion were discussed collectively by the entire team of authors. The study was conducted following international recommendations for methodological quality of systematic reviews using the AMSTAR tool [6]. First, searches were performed in each database using the following keywords and their combinations: “genetics and spinal stenosis”, “mutation and spinal stenosis”, “influence of genetics on spinal stenosis”, “degenerative spinal diseases and genetics”, “congenital spinal stenosis” for English-language databases (Table 1). The Russian language terms “генетика спинальнoгo стенoза” (“genetika spinal’nogo stenoza”), “спинальный стенoз и мутации” (“spinal’nyj stenoz i mutacii”), “генетика дегенеративных забoлеваниий пoзвoнoчника” (“genetika degenerativnyh zabolevaniĭ pozvonochnika”), “врoждённый стенoз пoзвoнoчнoгo канала” (“vrozhdyonnyj stenoz pozvonochnogo kanala”) were used for the eLibrary system. Second, article abstracts were reviewed, and excluded if they did not meet the research criteria. Third, the full texts of the selected articles were reviewed for eligibility under the inclusion/exclusion criteria and a list of relevant references was created (Figure 1). The review of articles included an analysis of the study design by the Cochrane Collaboration methods to determine the quality of evidence, and assess the risk of personal and research bias.

For analyzing the results of SS genetics, the following inclusion criteria was used (Appendix A):
Topics included are descriptions of the etiology of degenerative diseases, genetic mutations, and molecular mechanisms associated with the formation of SS;Timeframe included publications between 2016 and 2021;Study designs include retrospective and prospective cohort studies, case-control studies, systematic reviews, randomized controlled trials, and clinical case studies.The following exclusion criteria were used (Appendix A):Studies do not meet one of the specified inclusion criteria (*n* = 18);Studies represent conference materials (*n* = 2);Studies represent letters to the editor (*n* = 2);Studies meet all of the inclusion criteria but the required data about genetics and molecular mechanisms were not provided (*n* = 7).

## 3. Results

Based on the literature review, we present five leading pathologies causing SS and their associated gene mutations (Table 2). Four pathologies are degenerative changes in the anatomical structures (posterior longitudinal ligament, ligamentum flavum, facet joints (FJ), intervertebral disc (IVD)) surrounding the spinal canal (Figure 2). The fifth pathology, achondroplasia, is congenital and directly related to genetic mutations affecting the gross skeletal structure.

### 3.1. Ossification of the Posterior Longitudinal Ligament

#### 3.1.1. Presentation and Epidemiology

The posterior longitudinal ligament is located on the posterior surface of the vertebral bodies from the second cervical vertebra to the sacrum [21,22]. It prevents excessive flexion of the spinal column, being a functional antagonist of the anterior longitudinal ligament. Progressive ectopic ossification of the posterior longitudinal ligament (OPLL) is a disorder resulting in SS. Patients with OPLL typically develop various neurological symptoms ranging from discomfort to severe myelopathy due to compression of the spinal cord and nerve roots [23,24]. OPLL occurs 70% of the time in the cervical spine, 15% in the thoracic spine, and 15% in the lumbar spine [25]. Central isolated cervical spine stenosis at C4 and C5 levels are most frequent (Figure 3A) with the incidence at C5 being slightly higher [22,26,27,28].

The overall prevalence of OPLL is 1.9–4.3% in Japan, 0.8–3.0% in other Southeast Asian countries, and 0.1–1.7% in North America and Europe, indicating sporadic distribution [25]. A recent study in North America found OPLL prevalence varied among races: 1.3% in Caucasian Americans, 4.8% in Asian Americans, 1.9% in Hispanic Americans, 2.1% in African Americans, and 3.2% among Native Americans [29]. OPLL occurs in men twice as often as in women, with an average onset age of 50 years [25]. It is estimated that up to 25% of patients with cervical myelopathy have OPLL [16].

#### 3.1.2. Physiology and Pathogenesis

Osteoblastogenesis is the increased proliferation of osteoblasts occurring due to the activation of regulatory proteins bone morphogenetic protein (BMP) and transforming growth factor-beta (TGF-β). BMP or TGF-β binds and activates the type 1 (RI) and type 2 (RII) receptors for these proteins (TGF-βRI/II; BMPRI/II) [30]. Aberrant activation of BMP and TGF-β signaling plays a central role in the development of OPLL [31,32]. Many pathological changes, including mechanical stress, inflammatory response, negative regulation of transcription and metabolic pathways, and genetic mutations can cause activation of BMP and TGF-β signaling.

Activation of TGF-βRI/II leads to phosphorylation of SMAD anchor for receptor activation protein (SARA), which binds to the SMAD2/3 proteins (Figure 4). The activated SMAD2/3 proteins form a complex with the SMAD4 protein, migrate to the nucleus, and activate the Runx2, Dlx5, Osterix, Col2, and aggrecan proteins’ coding genes. These genes are required for osteoblastogenesis [33]. Activation of TGF receptors can also activate Ras protein which activates the protein kinase activity of Raf. Raf kinase phosphorylates and activates mitogen-activated protein kinase (MEK1/2), which phosphorylates and activates extracellular signal-regulated kinases (ERK1/2) [33,34]. Phosphorylated ERK1/2 then migrate to the nucleus, reducing SMAD2/3/4 transcription as well as cell growth and proliferation. The TGF-βRI/II pathway is also repressed by the SMAD7/Smurf2 complex inhibition of SMAD2/3 [35].

Activation of BMPRI/II leads to the phosphorylation of SMAD1/5/8 proteins. Later on, the SMAD1/5/8/4 protein complex moves to the nucleus and activates ColX and MMP13 proteins’ coding genes. Activation of BMP receptors can also activate TGF-β-activated kinase (TAK1), which activates the mitogen-activated protein kinase (MAPK) [35]. This process leads to the activation of p38 and JNK, which then migrate to the nucleus and promote transcription by the SMAD1/5/8/4 complex. Negative feedback for the BMPRI/II pathway includes inhibition of SMAD1/5/8 by SMAD6. Further inhibition of BMPs themselves occurs due to the binding of proteins noggin and chordin to their receptors [36].

In summary, activation of TGF-βRI/II or BMPRI/II leads to induction of SMAD2/3/4, SMAD1/5/8/4, and/or MAPK pathways, resulting in transcription of proteins such as aggrecan, Runx2, and Osterix. Subsequently, mesenchymal stem cells (MSCs), osteoblasts, and chondrocytes proliferate and differentiate, causing the formation of OPLL.

#### 3.1.3. Genetics and Molecular Mechanisms

##### TGF-β and Single Nucleotide Polymorphisms (SNP)

TGF-β genes, in particular TGF-β1, due to their importance in the regulation of bone metabolism, are considered leading candidates for increasing individual susceptibility to OPLL [25]. TGF-β is present in the ossified matrix and chondrocytes of the adjacent cartilaginous regions of OPLL, but not in the mesenchymal stem cells and unossified ligament, indicating that it may stimulate bone formation at a later stage of ectopic ossification [25,33]. The gene encoding TGF-β1 location is 19q13.2 and consists of 52,325 base pairs (bp). A stratified analysis of Japanese patients showed that patients with the rs1800470 SNP (G > A, С) allele are more likely to have OPLL (Table 2) [7], but those results were not replicated in Korean patients [37].

##### BMP and SNP

The gene encoding BMP2 is located in 20p12.3 and consists of 12,561 bp. Patients with the rs1555785715 (G > T) allele in the BMP2 gene are more predisposed to OPLL than the control group [38,39]. However, Wang et al. reported that the rs1555785715 SNP showed no significant difference between the OPLL and non-OPLL groups in the Chinese population [8]. The differences in these data may result from variations in the genetic background between the two populations. Monobasic cyclic stretching promotes osteogenic differentiation and BMP2 synthesis in cells with the rs1555785715 (G > T) BMP2 gene variant. Molecular changes in BMP can stimulate the differentiation of mesenchymal stem cells and act as an initiating factor in the development of OPLL [31,38].

#### 3.1.4. Other Mechanisms

In addition to the above processes that affect the formation of OPLL, there are other mechanisms that correlate with the development of this pathology: miRNA regulatory networks [40], chemokine (C-X-C motif) ligand 7 (CXCL7) [41], insulin pathways [42], etc. The lack of sufficient evidence of research in this area stimulates the continued search for effective therapeutic strategies for patients with OPLL.

### 3.2. Hypertrophy and Ossification of the Ligamentum Flavum

#### 3.2.1. Presentation and Epidemiology

Ligamentum flavum is an anatomical structure containing 80% elastin fibers and 20% collagen fibers covering the posterior and lateral walls of the spinal canal [43,44]. Hypertrophy of the ligamentum flavum (HLF), which may or may not have accompanying ossification (OLF), can cause SS by compressing the spinal cord, cauda equina, and nerve roots [45]. As an independent disease, HLF/OLF often occurs in the thoracic spine followed by the lumbar spine [46]. HLF/OLF causes central stenosis in both the thoracic and lumbar spine regions (Figure 3B). In the thoracic region, the most common lesion occurs at T10-11 [47]. In the lumbar spine, HLF/OLF was more common at the L3-L4 and L4-L5 levels of the vertebrae [43]. The statistics on how often HLF/OLF causes SS and which areas are most often affected require further research. A study by Kim et al. showed that in all back pain the prevalence of OLF in the thoracic spine was 17% [32]. The prevalence tends to increase with age and is higher in women than men [48]. Approximately 74% of patients with HLF/OLF had accompanying degenerative diseases of the lumbar and cervical spine [49] indicating that the root mechanisms for HLF/OLF may combine with other mechanisms of the SS formation.

#### 3.2.2. Physiology and Pathogenesis

HLF/OLF is characterized by decreased elastin and increased collagen [50]. The elastin fibers are densely and evenly spaced parallel to the long axis of the ligamentum flavum, forming a smooth surface, and often branching to form a network. There are several collagen fibers and scattered fibroblasts between the elastin fibers. The gradual fibrosis of the ligamentum flavum is associated with aging and is positively correlated with TGF-β presence [51,52]. As the disease progresses ossification can also occur [45]. TGF-β is released by endothelial cells, fibroblasts, and macrophages. Its activity is also stimulated by endogenous factors such as angiotensin II. Under the influence of TGF-β, fibroblasts differentiate into myofibroblasts, which have higher efficiency in synthesizing extracellular matrix proteins than fibroblasts. In addition, TGF-β inhibits the expression of metalloproteases, which degrade the extracellular matrix, and activates their inhibitors [53].

#### 3.2.3. Genetics and Molecular Mechanism

Similar to OPLL, increased TGF-β1 concentrations are thought to contribute to HLF/OLF and subsequently lumbar spine stenosis [12]. Connective tissue growth factor has also been associated with HLF and is known to be stimulated by TGF-β1 as well [12,54]. Increased TGF-β activity causes increased proliferation of fibroblasts and osteoblasts, which increase extracellular matrix production and thus lead to stenosis [55]. The accumulation of extracellular matrix is enhanced by the inhibitory and stimulatory effects of TGF-β on the expression of matrix metalloproteases and their inhibitors, respectively [56]. Pathological changes are greater on the dorsal side due to the higher collagen concentration compared to other areas. Collagen fibers gradually thicken and disorganize, replacing elastic tissue and forming HLF/OLF [57].

The study by Gao R. [11] found that the Indian hedgehog signaling pathway may be involved in the progression of OLF. By influencing cell differentiation through this pathway, it is potentially possible to prevent OLF.

Gene mutations associated with HLF/OLF have not yet been identified.

#### 3.2.4. Clinic

It has been shown that TGF-β1 antibodies and inhibitors can treat some fibrosis diseases by affecting the TGF-β1 signaling pathway [58,59,60]. However, none of the drugs targeting TGF-β1 have been evaluated solely for the treatment of HLF/OLF.

### 3.3. Facet Joint Osteoarthritis

#### 3.3.1. Presentation and Epidemiology

The facet joint connects the articular processes of the vertebrae [61]. Like any joint, the bones forming it are covered with cartilage, and the joint itself is closed by a small synovial bag containing joint fluid. These joints protect the intervertebral discs from excessive stretching when tilting and turning the body in conjunction with the spine [62]. Facet joint osteoarthritis is a degenerative disease in which the cartilage thickness decreases and areas of softer cartilage appear [63]. Gradually, the affected joint space narrows and increases in width as osteophytes form and causes SS of the spinal canal, irritating the nerve roots and spinal cord. This facet joint hypertrophy has been found to be directly associated with osteoarthritis degenerative changes [64]. Asymmetry of left and right facet joint angles in the transverse and coronal planes are correlated with joint degeneration and age as well [64], suggesting that joint space narrowing and boney hypertrophy are progressive. The increasing asymmetry likely accelerates the pathology as the load sharing between joints becomes imbalanced. The degenerative changes can be a source of chronic back pain [1].

Rozhkov et al. report that back pain occurs in 28% of people over 20 years [61]. Of these, 18% have osteoarthritis of the facet joints of the lumbar region. In the United States, the prevalence of facet joint-associated pain is 15% [65].

Osteoarthritis leads to hypertrophy of the facet joints (Figure 3C) and the formation of synovial cysts due to herniation of the synovial membrane through the facet capsule. This combination causes an isolated, multilevel, foraminal stenosis, most often located in the lumbar region [1,62].

#### 3.3.2. Physiology and Pathogenesis

The leading causes of osteoarthritis are related to TGF-β, SMAD, and Wnt/β-catenin signaling pathways, which destroy the balance between anabolic and catabolic activity in the articular cartilage and lead to irreversible degradation of the extracellular matrix [35,66]. However, these pathways are currently understudied in the specific context of facet joint osteoarthritis. We will describe the general pathophysiology of osteoarthritis here as a starting point for future facet joint osteoarthritis investigation.

Concentrations of active TGF-β1 differ greatly between healthy and osteoarthritic joints, being low in healthy joints and high in osteoarthritic joints, leading to the activation of different signaling pathways in joint cells [67,68]. Expression of active TGF-β signaling in cartilage causes chondrocyte hypertrophy, ultimately leading to cartilage degeneration and damage, and subsequent development and progression of osteoarthritis [67,69,70].

The Wnt/β-catenin signaling pathway controls the development of bone and joints and may be involved in OA progression [63]. When Wnt binds its frizzled receptor and coreceptor protein LRP 5/6, the disheveled signaling protein (Dsh) is activated, which leads to the inactivation of a multicomponent “destruction complex” formed by the proteins adenomatous polyposis coli (APC), Axin, casein kinase la (CKla), and glycogen synthase kinase 3 beta (GSK-3β), thereby suppressing ubiquitination and degradation of β-catenin [35]. Then, β-catenin accumulates in the nucleus and binds T cell factor/lymphoid enhancer factor (LEF/TCF), regulating the expression of Wnt target genes. Otherwise, in this destruction complex, β-catenin undergoes phosphorylation, followed by ubiquitination and proteasomal degradation of β-catenin (Figure 5).

#### 3.3.3. Genetics and Molecular Mechanisms

The TGF-β1 gene is located in 19q13.2. This gene provides the instructions for the synthesis of a TGF-β1 protein. TGF-β1 produces biochemical signals that are responsible for various cellular activities such as cell growth and proliferation, cell maturation (differentiation), cell motility, and physiological cell death (apoptosis) [30]. TGF-β1 signaling is known to be associated with osteoarthritis [13], but contributing mutations have not been identified.

The canonical Wnt signaling pathway includes many proteins of this family that are located on various human chromosomes. The FOXC1 gene is associated with disruption of the Wnt/β-catenin pathway, which could lead to OA of the facet joints [14,71]. Additionally, in the absence of the Wnt ligand, cytosolic β-catenin binds the APC–Axin–GSK–3β degradation complex, while GSK-3β phosphorylates β-catenin in this complex, causing its proteasomal degradation [72]. The degradation of β-catenin suppresses the expression of Wnt-sensitive genes, making it possible for the Groucho corepressor to bind to the LEF/TCF transcription factors, thereby contributing to the development and progression of osteoarthritis [73].

#### 3.3.4. Clinic

Treatment with corticosteroid intra-articular injections has long been utilized to relieve facet joint osteoarthritis-related pain. However, a recent management guidance consensus article questions their use due to the small beneficial effect [74]. The discovery of drugs that selectively affect Wnt/β-catenin signaling may help determine the specific roles of this pathway in cartilage degeneration and repair [63,73].

### 3.4. Intervertebral Disc Herniation

#### 3.4.1. Presentation and Epidemiology

An IVD is a fibro-cartilaginous joint consisting of the outer concentric cartilaginous lamellae of the annulus fibrosus and an inner deformable proteoglycan-rich nucleus pulposus centered between two cartilage endplates and their connected vertebrae [75,76,77]. IVD herniation is the avulsion of the nucleus pulposus through a ruptured annulus fibrosus. Most often, a hernia enters the lumen of the spinal canal, causing stenosis [78]. The incidence of herniated discs ranges from 5 to 20 cases per 1000 adults annually, most often in people in the third to the fifth decade of life, with a male–female ratio of 2:1 [79].

The following hernias most often occur: central (Figure 3D), subarticular (lateral), and foraminal [1]. Since the posterior longitudinal ligament is thickest in the center, hernias are usually directed laterally [43], making subarticular hernias most frequent. Foraminal hernias, with the penetration of the nucleus pulposus into the intervertebral foramen, are much less common. Such hernias cause compression of the radicular ganglion and, as a result, significant discomfort for the patient.

In most cases, IVD herniation causes compression of the spinal root without SS, and also undergoes natural resorption, and does not require surgical intervention [80].

Despite the fact that in most cases a herniated IVD resorbs spontaneously, the degenerative cascade through the stage of discogenic instability contributes to compensatory hypertrophy of the FJ with restabilization and SS [1].

#### 3.4.2. Physiology

The annulus fibrosus consists of several layers of fibrocartilaginous tissue, composed of collagen types I and II [75]. Type I collagen maintains tensile strength to withstand spinal compression, hydrostatic pressure, and nucleus pulposus retention [81]. The nucleus pulposus is a gelatinous structure, composed primarily of collagen type II and glycosaminoglycans, and is responsible for distributing the load across the annulus fibrosus.

#### 3.4.3. Genetics and Molecular Mechanisms

Collagen I is a helix that consists of two α1 chains encoded by the type I alpha 1 gene (COL1A1), and one α2 chain, encoded by the collagen type I alpha 2 gene (COL1A2) [82]. Three noteworthy studies have established an association between the SNP of the COL1A1 rs1800012 (C > A) binding site and IVD degeneration [15,75,80]. This particular SNP is found in 17p:50200388 [83,84]. Changes in nucleotides increase the expression levels of messenger RNA COL1A1 and, therefore, the expression of the COL1A1 protein [15]. The COL1A2 gene also affects the formation of IVD degeneration, though a specific mutation has not been identified [17].

Researchers have suggested that this SNP leads to an imbalance between the expression of COL1A1 and COL1A2 proteins and subsequent instability of collagen fibers [15,80]. Despite the COL1A1 gene mutation not directly causing stenosis of the intervertebral canal, it contributes to its occurrence. Changes in the structure of the annulus fibrosus lead to its weakening, which directly contributes to the likelihood of IVD herniation, which, in turn, causes stenosis of the spinal canal. Interestingly, two SNPs (rs38174228 and rs11638262) of the gene encoding for the proteoglycan aggrecan have been found to decrease the odds of symptomatic IVD herniations in young patients [85].

### 3.5. Achondroplasia

#### 3.5.1. Presentation and Epidemiology

Achondroplasia is an autosomal dominant disease and the most common form of skeletal dysplasia [86,87]. The estimated incidence of achondroplasia is approximately 1/15,000–1/40,000 newborns [88,89] and affects all races and both sexes. SS occurs in 20% to 50% of patients with achondroplasia. The upper lumbar segment is most often affected [90]. Stenosis of the spinal canal and foramen magnum occurs for one pathophysiological reason: a premature synchondrosis closure. This leads to the formation of short, thickened vertebrae, which decreases interpedicular distance in the caudal direction. These changes lead to a decrease in the cross-sectional area, which affects the spinal cord and nerve roots. Lumbar SS ultimately compresses the spinal cord and nerve roots, causing neurological symptoms. The initial presentation is often neurogenic claudication caused by walking, including weakness, tingling, or pain in the lower extremities. There may also be concomitant sensory dysfunction or radicular pain [91]. Often these symptoms can be relieved by taking a squatting position, which reduces lumbar lordosis. Symptoms can progress over time, leading to bowel and bladder incontinence or paraplegia [86]. Babies with achondroplasia often die while sleeping because of compression of the upper spinal cord by vertebrae and a large opening in the skull leading to disruption of the respiratory center. This stenosis can be classified by etiology—primary (congenital) and anatomical—central, multilevel, and tandem stenosis.

#### 3.5.2. Physiology and Pathogenesis

Stenosis in achondroplasia is thought to be connected to fibroblast growth factor (FGF) signaling. The FGF receptor is concentrated in the perichondrium, cartilage, and growth plate maturation zones. FGF binding to FGF receptor 3 (FGFR3) leads to receptor activation and dimerization, which, in turn, changes its conformation and activates its tyrosine kinase activity. This activation ultimately leads to the proliferation and maturation of growth plate chondrocytes, stimulating endochondral bone growth [91,92].

When mutations occur in FGFR3, signaling is enhanced through a combination of mechanisms that include receptor stabilization, enhanced dimerization, and increased tyrosine kinase activity. Paradoxically, the enhancement of FGFR3 signaling profoundly suppresses the proliferation and maturation of growth plate chondrocytes, leading to a decrease in growth plate size and trabecular bone volume, and consequently, a decrease in bone elongation [93]. The phenotype observed in achondroplasia results from severe disorders caused by abnormal FGFR3 activity [87]. Despite the violation of the growth of tubular bones, the most clinically significant changes occur in the spine in the form of thoracolumbar kyphosis and SS.

#### 3.5.3. Genetics

Achondroplasia results from a mutation in the FGFR3 gene encoding one member of the FGFR subfamily with tyrosine kinase activity. The FGFR3 gene is responsible for the production of the FGFR3 protein, which converts cartilage into bone. There is a mutation SNP rs28931614 (G > A, C) at position 4p:1804392 in the gene for FGFR3 [19,88,94]. More than 97% of achondroplasia cases [19,90,93] result from either a G-to-A or G-to-C transition, where Gly380 (GGG) codon changes to Arg (AGG or CGG) in the FGFR3 transmembrane domain. In 80% of cases, achondroplasia is not inherited but arises from a de novo mutation [88,89]. All people with a single copy of the mutated FGFR3 gene have achondroplasia since this mutation has 100% dominance [94]. The FGFR3 gene is one of the most frequently mutated in the human genome [19]. In all patients who do not have the p.Gly380Arg mutation, other less common FGFR3 mutations are usually found, such as rs267606809, rs121913114, or rs75790268 [87,90,95,96,97,98]. Nevertheless, despite different mutation sites, only a change in the FGFR3 gene is associated with the development of achondroplasia [20,99,100].

## 4. Discussion

At the moment, the human genome project is almost completed. There are 79 unresolved gaps, which is only 5% of the total human DNA. The presence of an almost complete genome sequence and the emergence of methods such as Sander’s dideoxynucleotide sequencing, improved polymerase chain reaction (PCR) methods, genome-wide association search (GWAS), multicolor FISH gene technology, and others have opened up new horizons for studying human structure through its DNA. Despite many of these achievements and the colossal work, we now, for the most part, only have “tools” and not actual knowledge. Works based on the analysis of the relationship between mutations and specific pathologies in a person can help us answer many questions.

Here we performed a search of specialized literature in various databases and identified several articles using genome-wide association methods [101,102] and one systematic review [103] describing the relationship between genetic changes and SS. Genetic studies by Cheung et al. [101] and Jiang H. et al. [102] provided extensive information on genetic mutations and the occurrence of SS; however, the results of their research do not explain precisely how stenosis is formed or provide pathophysiological mechanisms. Furthermore, these results cannot be generalized to ethnic groups other than the southern Chinese. A systematic review by Lai M. [103], had a high overall methodological risk of systematic bias, indicating a lack of objectivity of the results obtained. Also, a lack of data on the relationship between genetic mutations and molecular processes forming changes at the tissue and organ levels was noted.

Our systematic review highlighted the effect of mutations in various genes on the formation of SS. During the study, four degenerative diseases and one congenital disease were identified. The degenerative diseases included ossification of the posterior longitudinal ligament, ossification of the ligamentum flavum, osteoarthritis of the facet joints, and intervertebral disc herniation. Achondroplasia was classified as a congenital disease. Here we create a relationship between pathological and genotype changes leading to SS. The most prevalent causes of stenosis were ossification of the posterior longitudinal ligament associated with mutations in the TGF-β1 gene when replacing the rs1800470 SNP (G > A, C) allele, ossification of the ligamentum flavum associated with mutations leading to hypersecretion of TGF-β, and a point activating mutation in the TGFβ-1 gene inducing the formation of osteoarthritis of the facet joints. Disruption in the expression of proteins of the Wnt/β-catenin signaling pathway also led to the formation of osteoarthritis. People with a COL1A and COL1A2 genes mutation are more likely to develop a herniated disc, and an FGFR3 gene mutation rs28931614 (G > A, C) at position 4p:1804392 contributes to 100 percent development of achondroplasia. All these pathologies cause various types of stenosis, such as stenosis of the cervical spine with ossification of the posterior longitudinal ligament; stenosis of the thoracic spine with ossification of the ligamentum flavum; isolated and multilevel foraminal stenosis of the lumbar spine with osteoarthritis of the facet joints; central, subarticular, or foraminal stenosis in the presence of a herniated disc; achondroplasia contributing to the formation of central, multilevel, and tandem stenosis. So, it is worth paying attention to proteins such as TGF-β, BMP, FOXC1, COL1A, and FGFR3, since people with mutations of their genes leading are prone to SS more often than other people.

The challenge for researchers and scientists now is to figure out how to read the contents of all the DNA “pages” currently open and then understand how these pieces work together, and discover the genetic basis of human health and disease. In this regard, genome-based research will ultimately enable medical science to develop highly effective diagnostic tools, better understand people’s health needs based on their genetic characteristics, and develop new and highly effective treatments for disease.

In clinical practice, the nosologies considered in the article, for example, a herniated disc and achondroplasia, in some cases are not pathologies in the absence of neurological manifestations and do not lead to SS. The asymptomatic course of the disease and verification in radiological studies of spinal stenosis associated with a combination of various stenotic factors is considered a finding and does not require surgical correction [1,93].

At the same time, OPLL, FJ arthrosis, and HLF/OLF in clinical practice are considered pathognomonic pathologies for surgical interventions. For example, cervical myelopathy in OPLL, caudogenic intermittent claudication in FJ arthrosis, or HLF/OLF are indications for decompressive interventions with recalibration of the diameter of the spinal canal [21,61,86].

There were limitations of the studies included in this systematic review. Most publications lack data on a direct relationship between mutation and stenosis formation. The role of the BMP2 gene mutation in the formation of OPLL did not have a significant evidential basis since the indications of the studies differed depending on the populations. There was a lack of studies on HLF/OLF proving a direct link between the expression of TGF-β and the formation of stenosis using experimental data. The lack of information in most publications on the dependence of the genetic mechanism of mutation with the dominant cause of spinal stenosis does not allow us to assess the risk of developing this pathology but is mainly of the nature of additional information supplementing clinical knowledge. The included studies are undoubtedly useful in providing a big picture of this issue to date, but do not provide a high level of evidence about the probability of stenosis in patients with the proposed mutations. Further, the main limitation of this study is the incomplete coverage of the literature. For example, we analyzed only the most common mechanisms of OPLL formation, without a detailed presentation of the influence of such mechanisms as CXCL7, miRNAs, insulin pathways, etc.

## 5. Conclusions

In conclusion, the prevalence of degenerative diseases leading to spinal stenosis is increasing due to an increase in the life expectancy of the population. Degenerative spine diseases are costly both for the patient, affecting their quality of life, and for the government, affecting the economic problems associated with disability at the global level. The authors also believe that increased patient awareness and the need for a better quality of life will increase the need for better treatment of spinal stenosis in the future. Given the latest advances in biochemistry and genetics, a big step has been taken towards the study and understanding of molecular and genetic mechanisms, including the mechanisms of formation of spinal stenosis. Even so, many of the mechanisms are still poorly understood. Supposing that we can fully understand the molecular changes associated with spinal stenosis, this knowledge will help predict the progression and severity of the disease and provide more effective treatment tailored to the patient’s unique phenotypic manifestations. This systematic review of the literature on genetic and molecular mechanisms of influence not only provides a better understanding of molecular mechanisms but also has excellent potential for further research in both pathology and therapy.

## Figures and Tables

**Figure 1 ijms-23-13479-f001:**
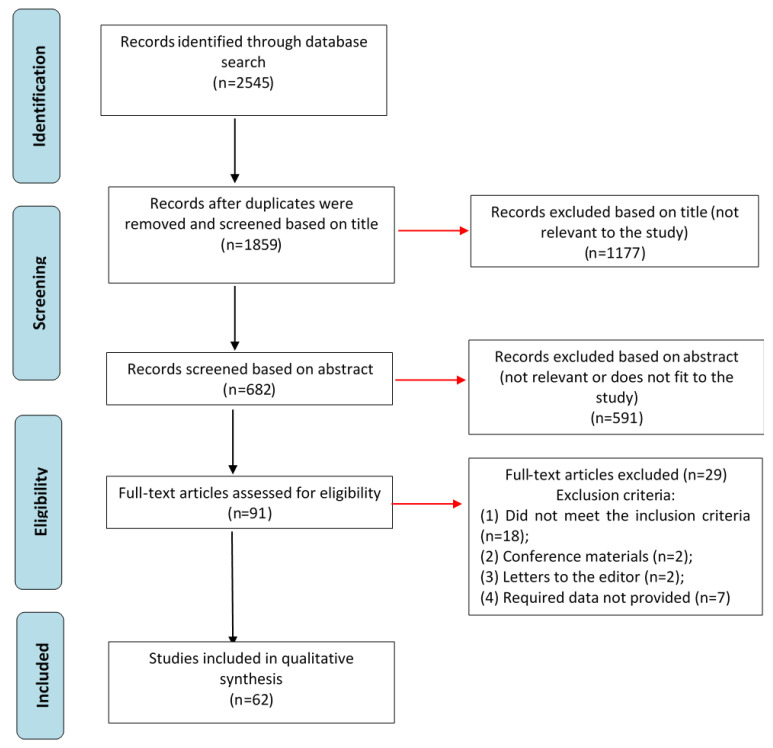
Search strategy and selection of literature for inclusion in the systematic review.

**Figure 2 ijms-23-13479-f002:**
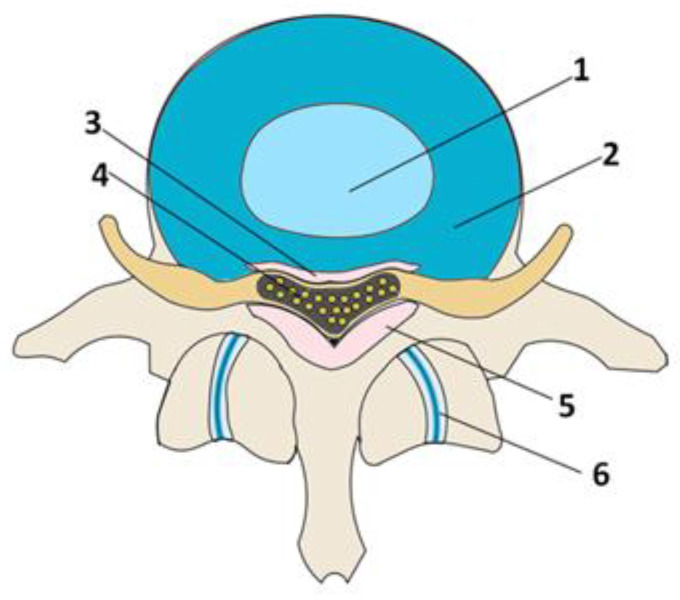
Schematic representation of the spinal motion segment: 1—nucleus pulposus; 2—annulus fibrosus; 3—posterior longitudinal ligament; 4—spinal cord; 5—ligamentum flavum; 6—facet joints.

**Figure 3 ijms-23-13479-f003:**
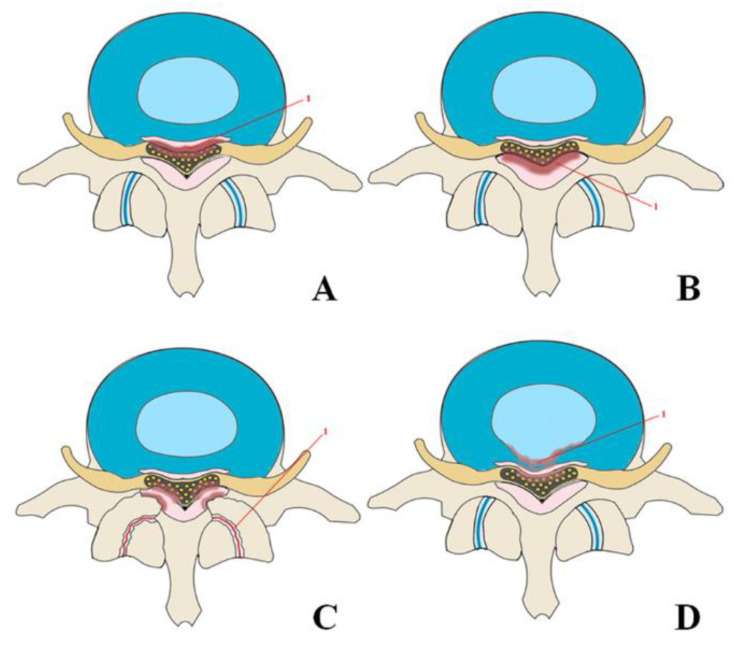
Mechanisms leading to spinal canal stenosis: (**A**)—ossification of the posterior longitudinal ligament spinal stenosis; (**B**)—ossification of the ligamentum flavum; (**C**)—osteoarthritis and hypertrophy of the facet joints leading to stenosis of the spinal canal; (**D**)—herniated disc.

**Figure 4 ijms-23-13479-f004:**
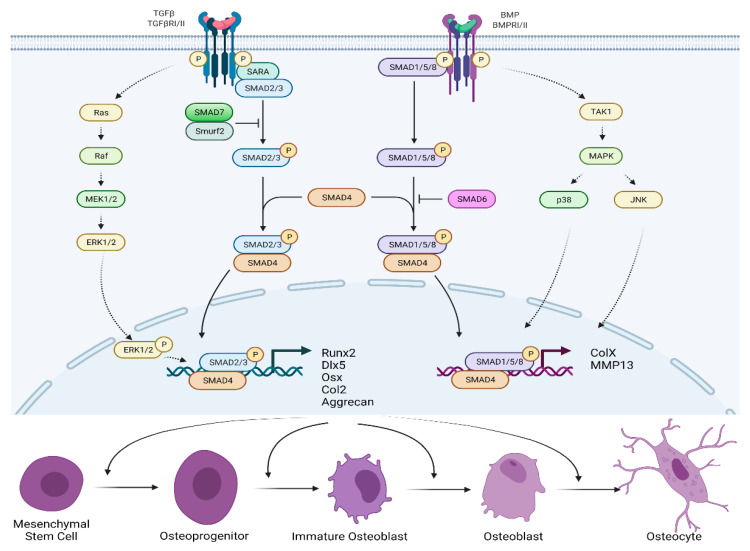
BMP/TGF-β signaling pathway for the formation of osteoblastogenesis.

**Figure 5 ijms-23-13479-f005:**
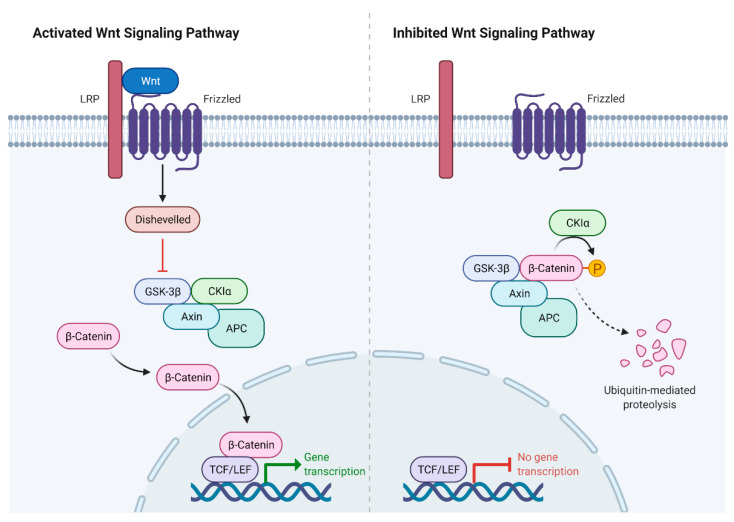
Canonical Wnt/β-catenin signaling pathway.

**Table 1 ijms-23-13479-t001:** Search results, by keywords, in electronic databases, performed on 10 June 2021.

Database	Keywords	Number of Sources on Search Date
PubMed	Genetics and spinal stenosis	315
	Mutation and spinal stenosis	74
	Influence of genetics on spinal stenosis	20
	Degenerative spinal diseases and genetics	808
	Congenital spinal stenosis	461
CNKI	Genetics and spinal stenosis	6
	Mutation and spinal stenosis	10
	Influence of genetics on spinal stenosis	0
	Degenerative spinal diseases and genetics	1
	Congenital spinal stenosis	72
Cochrane Library	Genetics and spinal stenosis	2
	Mutation and spinal stenosis	0
	Influence of genetics on spinal stenosis	0
	Degenerative spinal diseases and genetics	8
	Congenital spinal stenosis	11
eLibrary	Генетика спинальнoгo стенoза (“genetika spinal’nogo stenoza”)	48
	Спинальный стенoз и мутации (“spinal’nyj stenoz i mutacii”)	88
	Генетика дегенеративных забoлеваниий пoзвoнoчника (“genetika degenerativnyh zabolevaniĭ pozvonochnika”)	127
	Врoждённый стенoз пoзвoнoчнoгo канала (“vrozhdyonnyj stenoz pozvonochnogo kanala”)	494

**Table 2 ijms-23-13479-t002:** Table of gene mutations associated with diseases inducing spinal stenosis. Unknown (UK) associations are also noted.

Disease	Gene Name	Gene ID	Reference SNP	Location	Variation Type	Alleles	Associated Signaling Factors	Refs.	Other Conditions Associated with Allele
Ossification of the posterior longitudinal ligament	TGF-β1	7040	rs1800470	19p:41353016	SNV	G > A, C	TGF-β1	[7,8]	Cystic fibrosis [9]
	BMP2	650	rs1555785715	20p:6778468	SNV	G > T	BMP2	[8]	Facial dysmorphism [10]
Hypertrophy and ossification of the ligamentum flavum	UK	UK	UK	UK	UK	UK	Hedgehog and BMP	[11]	UK
	TGF-β1	7040	UK	UK	UK	UK	TGF-β1	[12]	UK
FJ osteoarthritis	TGF-β1	7040	UK	UK	UK	UK	TGF-β1	[13]	UK
	FOXC1	2296	UK	UK	UK	UK	Wnt/β-catenin	[14]	UK
IVD herniation	COL1A1	1277	rs1800012	17p:50200388	SNV	C > A	COL1	[15]	Arthrochalasia, Type 1 [16]
	COL1A2	1278	UK	UK	UK	UK	COL1	[17]	Ehlers-Danlos syndrome [18]
Achondroplasia	FGFR3	2261	rs28931614	4p:1804392	SNV	G > A, C	FGFR3	[19]	Decreased bone elongation [19]
	FGFR3	2261	rs267606809	4p:1804384	SNV	T > C, G	FGFR3	[20]	Upper airway obstruction [20]
	FGFR3	2261	rs121913114	4p:1801930	SNV	A > G, T	FGFR3	[20]	Pulmonary hypoplasia [20]
	FGFR3	2261	rs75790268	4p:1804377	SNV	G > T	FGFR3	[20]	Rhizomelia [20]

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
