# Peer review of "Molecular and Genetic Mechanisms of Spinal Stenosis Formation: Systematic Review"

_ijms, 2022, doi:10.3390/ijms232113479_

Round 1

Reviewer 1 Report

Dear colleagues.

Thanks for your attempt on this important subject. The review was well-performed within the parameters you described.

My main concern is that there are tenuous links between the main question - clinical spinal stenosis - and the molecular mechanisms described. You acknowledge as such in the limitations section.

A discussion of the effect-size of the mutations you discuss would be important - in your estimation, are the majority of SS cases associated with a particular genetic variant? Is it a strong or weak effect (RRs or ORs could be mentioned)? You do acknowledge that some of the findings have not been replicated. Are these generalizable?

In addition, one could argue that disc herniations, OPLL and achondroplasia are not really conditions we consider as SS.

Disc herniations are perhaps a very distinct phenomenon with a very different natural history - most improve spontaneously. As such, herniations mostly don't result in SS.

OPLL has been the subject of many reviews, and some of the mechanisms recently proposed are not discussed (CXCL7, miRNAs, insulin pathways, etc)

Finally, in clinical practice, ligamentum flavum hypertrophy along with facet joint hypertrophy seems to be very important / pathognomonic. This is not at all discussed here.

In sum, I think your review is a nice effort, but misses the mark in my opinion.

Author Response

Dear Reviewer 1,

We would like to thank you and the reviewers for their insightful comments and suggestions. We will address each of their suggestions on a point-by-point basis. All of the changes to the manuscript are colored green in Word.

We thank the reviewer for the kind comments.

Reviewer 2 Report

Dear authors,

the review article is well researched and clearly written with some minor issues. 

In line 150 you state that cells "differentiate and proliferate". Differentiation and proliferation are mutually exclusive with proliferation usually preceding the differentiation process (transient amplifying progenitors), therefore I would suggest "proliferate and differentiate"

In line 179 there is a typo. The term should say "cauda equina" not "cauda equine"

In line 188 you state that the "root mechanisms for OLF may overlap with other causes".  It is unclear to me what you intend to say as I'm not sure how a mechanism and a cause can formally overlap. Please rephrase the sentence.

In line 264 you introduce Wnt signaling. While, there is some controversy about the convergence of the Wnt signaling pathways, most experts would refer to the pathway that you depict "canonical Wnt signaling" as opposed to "non canonical Wnt signaling"

In line 422-423 you state that the main limitation of the study is the incomplete coverage of the literature. Yet in 437 you offer "This comprehensive review...". That is contradictory.   Maybe use "systematic review" as you offer in the title of the manuscript.

Please transliterate the cyrillic terms and provide an english translation. (57-60) 

Figures 2 and 3 are nicely done but the numbers for the structures are practically unreadable (font size) when printed out.

Author Response

Dear Reviewer 2,

We would like to thank you and the reviewers for their insightful comments and suggestions. We will address each of their suggestions on a point-by-point basis. All of the changes to the manuscript are colored green in Word.

We thank the reviewer for the kind comments.

Round 2

Reviewer 1 Report

Dear Authors,

Thanks for your submission. Unfortunately, my initial concerns stand - the dominant cause for spinal stenosis is degenerative disease with facet hypertrophy and ligamentum hypertrophy - and that is not at all addressed here. Even though facet hypertrophy is discussed in a general sense, the references are general and pertain more to osteoarthritis / cartilage homeostasis - not necessarily facet disease.

The references to Camurati-Englmann disease have very little to do with common spinal stenosis. It is a major stretch to associate this condition with facet joint OA.  

There is nothing discussing ligamentum hypertrophy - just OLF - not the same in my view.

The comments from my first review remain - despite the authors' attempts at an answer

Author Response

Dear Academic Editor and Reviewer,

We would like to thank Editor and the reviewer for insightful comments and suggestions. We will address each of their suggestions on a point-by-point basis. All of the changes to the manuscript are colored green in Word. All reference numbers were updated as well.
